# Papillary Thyroid Carcinoma with Spindle Cell Metaplasia: A Rare Encounter

**DOI:** 10.3390/diagnostics12040855

**Published:** 2022-03-30

**Authors:** Ka Wen Leong, Shahrun Niza Abdullah Suhaimi, Geok Chin Tan, Yin Ping Wong

**Affiliations:** 1Department of Pathology, Faculty of Medicine, Universiti Kebangsaan Malaysia, Kuala Lumpur 56000, Malaysia; kawen_leong@yahoo.com; 2Department of Surgery, Faculty of Medicine, Universiti Kebangsaan Malaysia, Kuala Lumpur 56000, Malaysia; shahrun72.sn@gmail.com

**Keywords:** epithelial-mesenchymal transition, metaplasia, papillary carcinoma, spindle cell, thyroid, variant

## Abstract

A myriad of histological variants of papillary thyroid carcinoma (PTC) have been described, some of which can be diagnostically challenging due to their rarity and overlapping histomorphology with other entities. One of the scarce and poorly characterised variants is PTC with spindle cell metaplasia, of which fewer than 20 cases have been reported in the literature hitherto. Our patient was a 51-year-old woman with a four-month history of painless, gradually enlarging neck swelling. Physical examination revealed a solitary left thyroid nodule. Thyroid ultrasonography demonstrated a hypoechoic nodule with irregular borders and speckles of microcalcification at the periphery. Total thyroidectomy with central and lateral lymph node dissection was performed. Grossly, there was a poorly circumscribed mass occupying the entire left thyroid lobe measuring 30 mm in the largest dimension. Histopathological examination revealed features of a classical PTC. Incidentally, a well-circumscribed 9 mm nodule was identified within the tumour mass. The nodule comprised of spindle cells arranged in loose fascicles, displaying uniform bland looking nuclei. No mitosis, necrosis or nuclear atypia was observed. Immunohistochemically, the spindle cells were immunopositive to TTF-1 and thyroglobulin, indicating thyroid follicular cell lineage. p53 and BRAF V600E mutant protein immunoexpression were focally noted. They were negative for calcitonin, S100, and desmin. Loss of E-cadherin and CK19 were also demonstrated. A diagnosis of PTC with spindle cell metaplasia was rendered. The nature of spindle cell in PTC needs to be meticulously defined. Careful histomorphology examination and judicious use of immunohistochemistry stains are helpful in arriving at an accurate diagnosis.

## 1. Introduction

Papillary thyroid carcinoma (PTC) is the most common histological subtype of thyroid carcinoma, accounting for approximately 80% of cases. It is characterised histologically by the presence of malignant epithelial proliferation harbouring typical nuclear changes such as overlapping ground-glass and grooved nuclei with intranuclear cytoplasmic pseudoinclusions [1].

A myriad of histological variants of PTC have been described, some of which can be diagnostically challenging due to their rarity and overlapping histomorphology with other entities. One of the scarce and poorly characterised variants is PTC with spindle cell metaplasia, of which less than 20 cases have been reported in the literature hitherto, primarily as isolated case reports [2,3,4,5,6]. Notably, the term PTC with spindle cell metaplasia or spindle cell variant of PTC is used interchangeably [1]. It has distinctive plump to thin and elongated spindle cells arranged in a well-circumscribed nodule, constitute from 1% to almost 100% of the entire tumour bulk. Although spindle cells are not a common feature encountered in thyroid pathology, it was previously described in association with some reactive, hyperplastic, and neoplastic processes, including post-fine needle aspiration spindle cell nodule of the thyroid, multinodular goitre, follicular adenoma, and a subset of PTC [7].

This report aimed to shed more light on this rare variant of PTC with a review of the literature on morphological identification, clinical characteristics, and prognosis. The histomorphology and immunohistochemistry features that are helpful to arrive at an accurate diagnosis were also discussed.

## 2. Case Presentation

### 2.1. Clinical Presentation

A 51-year-old woman presented to us with a four-month history of painless gradually enlarging anterior neck swelling. There were neither symptoms related to airway obstruction nor hyper-/hypothyroidism. Physical examination revealed a solitary, firm left thyroid nodule with an irregular surface measuring 25 mm in the largest dimension. It was associated with multiple smallipsilateral cervical lymphadenopathy. Ultrasonography of the thyroid demonstrated a hypoechoic nodule with irregular borders measuring 16 × 13 mm over the left lobe. Speckles of microcalcification were seen at the periphery (Figure 1). The rest of the thyroid was unremarkable. A few small lymph nodes with radiological features suspicious of metastasis (hyperechogenicity and loss of fatty hilum) were seen over the ipsilateral lateral group.

### 2.2. Cytopathological and Histopathological Examination

Fine needle aspiration of the thyroid nodule was performed, showing cellular smears comprised of loose cohesive sheets of neoplastic thyroid follicular cells arranged in papillary configuration. The neoplastic thyroid follicular cells exhibited enlarged, overlapping nuclei with occasional nuclear grooves and intranuclear cytoplasmic pseudoinclusions. Cytological features were consistent with a classical PTC (Bethesda diagnostic category 6). Total thyroidectomy with central and left lateral level III and IV neck dissection was carried out based on the cytological diagnosis of a PTC and the presence of suspicious lymph nodes in the ipsilateral lateral group.

Grossly, a poorly circumscribed tumour mass occupied almost the entire left lobe measuring 30 mm in largest dimension, with focal breaching of capsule seen at the upper pole. The right thyroid lobe and isthmus were unremarkable. Histopathological examination of the tumour mass revealed features of a classical PTC, comprised of neoplastic follicular cells arranged in complex, arborizing, and closely packed papillary architectures. The neoplastic follicular cells showed enlarged, overlapping nuclei with finely dispersed optically clear chromatin, readily seen nuclear grooves, and intranuclear cytoplasmic pseudoinclusions.

Adjacent to the tumour mass, there was an incidental finding of a well-circumscribed nodule composed predominantly of spindle cells in fascicles devoid of papillary architecture, measuring approximately 9 mm in greatest dimension (Figure 2A,B). The spindle cells displayed uniform, plump to thin elongated nuclei with relatively abundant pale eosinophilic cytoplasm. No nuclear grooves and intranuclear cytoplasmic pseudoinclusions were found. No increase in mitosis nor necrosis was noted (Figure 2C). Entrapped tumour cells within the nodule exhibiting features consistent with classical PTC were also observed (Figure 2D). Section from the cervical lymph nodes showed evidence of metastasis from the classical PTC.

The spindle cells were immunoreactive toward thyroid transcription factor-1 (TTF-1) and thyroglobulin, indicating thyroid follicular cell lineage. They were also immunopositive for vimentin, CKAE1/AE3, and were found focally expressed p53 and BRAF V600E mutant protein. They were negative for calcitonin, chromogranin, synaptophysin, S100, smooth muscle actin (SMA), and desmin. Loss of E-cadherin and CK19 were demonstrated in the spindle cells, while the entrapped tumour cells of classical PTC remained positive (Figure 3). The spindle cell component exhibited a low Ki67 proliferative index, approximately 1% in comparison with the follicular cell counterpart.

A diagnosis of PTC with spindle cell metaplasia with cervical nodal metastasis was rendered. The tumour was staged as pT2N1b based on The American Joint Committee on Cancer (AJCC) staging system. The patient was subsequently treated with radioactive iodine and showed a good recovery. There was no recurrence at twelve months follow-up.

## 3. Discussion

Since the first description of spindle cell metaplasia in papillary carcinoma of thyroid by Hutter et al. in 1965 [8], spindle cell proliferation has been reported in a myriad of thyroid lesions including both the reactive, benign and malignant thyroid neoplasms. Spindle cell metaplasia arising from PTC is exceedingly rare, with fewer than 20 cases having been described in the English literatures hitherto [2,3,4,5,6].

According to the literature, PTC with spindle cell metaplasia shows a female preponderance with a female to male ratio of 9:4. It occurs in adult patients in an age range from 25 to 67 years old. The lesions vary in size and range from 3 mm up to 50 mm [2,3,4,5,6]. There is no specific laterality of the lesion. It is commonly described in association with classical and follicular variants of PTC. Histologically, the spindle cells exhibit plump to slender, deceptively bland-looking nuclei with finely granular chromatin and inconspicuous nucleoli, and ample eosinophilic cytoplasm with an indistinct cell border. Intriguingly, the unique nuclear features of classical PTC are rarely observed [2]. The spindle cell component constitutes from 1% to almost 100% of the entire tumour bulk. It can appear in a multicentric, nodular or diffuse infiltrative growth pattern. Often, there are entrapped isolated PTC follicles seen, as the present case. Ki67 proliferative index of the spindle cell component is low (less than 2%). The prognosis of this rare entity remains poorly characterised, but likely portrays a promising outcome with a standard treatment regimen [3].

Table 1 illustrated the clinicopathological features of all the 13 reported cases (including one of ours) in the English literatures. The immunohistochemistry profile of this unique entity is also summarised. Briefly, the spindle cells are immunoreactive towards thyroglobulin, TTF-1, vimentin and cytokeratin, and were negative for SMA, desmin, calcitonin, S100, epithelial membrane antigen (EMA), chromogranin and synaptophysin [2,3,4,5,6]. The immunohistochemical profile suggests that these spindle cells originated from thyroid follicular epithelial cells. Interestingly, these spindle cells showed focal p53 and BRAF V600E immunopositivity, a known mutant protein in papillary thyroid carcinoma [9].

The term “metaplasia” is defined as a reversible change from one differentiated cell type to another mature differentiated cell type in the context of abnormal stimulus or environmental stress. Metaplastic change in malignant cells is unusual. The precise mechanisms of spindle cell metaplasia in PTC remain debatable and requires further studies. Herrmann and Trevor (1993) demonstrated, in one of their in vitro studies on thyroid tumour cells, the transition from differentiated epithelial cells to mesenchymal cells, suggesting the unique nature of thyroid follicular cells and the role of epithelial-mesenchymal transition (EMT) in tumourigenesis [10]. The trigger for metaplastic transformation of thyroid follicular cells might be attributed to the cell-matrix interactions during neoplastic growth or under the influence of cytokine produced by the follicular neoplastic cells [10,11]. In addition, the loss of E-cadherin and CK19 immunoreactivity in the spindle cells of the present case, and in other previous studies [2,3], which have suggested that it could be a mesenchymal-like metaplasia in the follicular cell elements.

### Histological Differential Diagnosis of Thyroid Lesions with Spindle Cell Morphology

The spindle cells in this entity can be deceptively bland, indistinguishable from a reactive spindle cell lesion of the thyroid, such as post-procedure spindle cell nodules (PSCN) [12]. The lesion typically occurs weeks to months after a traumatic procedure. PSCNs are usually non-encapsulated, encompass bland-looking spindle cells with accompanied delicate blood vessels, mixed inflammatory infiltrates and myxoid change within the stroma. In contrast to spindle cell metaplasia in PTC, they are myofibroblastic in origin and show SMA immunoreactivity [12].

Smooth muscle tumours of the thyroid, although rare, should be considered as one of the differential diagnoses when dealing with spindle cell lesions of the thyroid [13]. Leiomyoma is characteristically well-circumscribed with uniform, cigar-shaped nuclei arranged in interlacing bundles, mimicking spindle cell metaplasia in PTC to perfection. Muscle immunomarkers such as SMA, h-caldesmon and desmin are helpful to highlight the muscular differentiation in the former [13].

Besides, peripheral nerve sheath tumours (PNST), including neurofibroma and schwannoma of the thyroid, can sometimes be confused with spindle cell metaplasia in PTC, especially when the classical verocay bodies of schwannoma are not obvious in the biopsy. Histologically, PNST has thin, wavy, spindle-shaped nuclei with tapering ends. Immunohistochemically, they show S100 immunopositivity and are negative for thyroid markers such as thyroglobulin and TTF-1 [14].

Spindle epithelial tumours with thymus-like differentiation (SETTLE) is a rare tumour of the thyroid with metastatic potential. It can be indistinguishable histologically from spindle cell metaplasia of PTC with the presence of biphasic spindle and epithelial elements [15]. The immunopositivity of the spindle cells for thyroglobulin and keratins indicated a follicular cell lineage (in the present case) and excludes the possibility of SETTLE.

Medullary thyroid carcinoma (MTC) is a neuroendocrine tumour arising from the parafollicular C cells of the thyroid gland, and constitutes less than 10% of malignant thyroid neoplasms. Spindle cell variant of the MTC may sometimes resemble spindle cell metaplasia in PTC. The diagnosis of the former, however, can be supported by the demonstration of amyloid deposits and calcitonin immunoreactivity [16].

Anaplastic thyroid carcinoma (ATC), as the most aggressive malignant tumour of the thyroid, is typically composed of bizarre spindle cells with multinucleated giant cells or osteoclast-like neoplastic cells. Mitosis is a feature of this [1]. Hutter et al. (1965) was the first to describe the occurrence of spindle cell metaplasia in association with PTC. Notably, these were large (>50 mm), rapidly enlarging, tumours with associated compressive symptoms with recurrent laryngeal nerve palsy, and portend a dismal prognosis [8]. These aggressive lesions might represent true anaplastic transformation of PTC instead. Similarly, Brandwein–Gensler et al. (2004) reported a case of spindle cell transformation of papillary carcinoma as a potentially lethal variant. Immunostains revealed that the spindle cells were immunoreactive for vimentin, but not with thyroglobulin and cytokeratin [17], which we believed could represent another unusual variant of anaplastic transformation of PTC.

An accurate recognition of this rare lesion is crucial due to a broad range of possible differential diagnoses, varying from reactive benign conditions to very aggressive life-threatening malignancies like ATC. Careful histomorphology examination and judicious use of immunohistochemistry stains are helpful in arriving at a correct diagnosis.

## 4. Conclusions

The nature of spindle cells in PTC is still poorly defined. Further studies are required to delineate the pathogenesis of its occurrence. This report described in detail the clinicopathological characteristics of this rare entity, which we believe will increase the knowledge of our existing understanding on the spindle cell association with PTC.

## Figures and Tables

**Figure 1 diagnostics-12-00855-f001:**
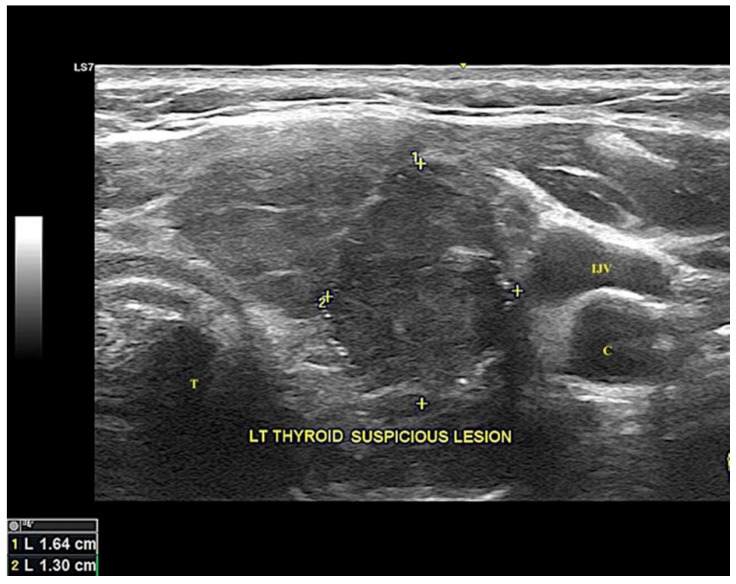
Ultrasonography image of the thyroid reveals a suspicious hypoechoic nodule with irregular border seen at the left thyroid lobe measuring 16.4 × 13.0 mm in diameter, outlined by “+”. Speckles of microcalcification are seen at the periphery. C = carotid artery, LT = left, IJV = internal jugular vein, T = trachea.

**Figure 2 diagnostics-12-00855-f002:**
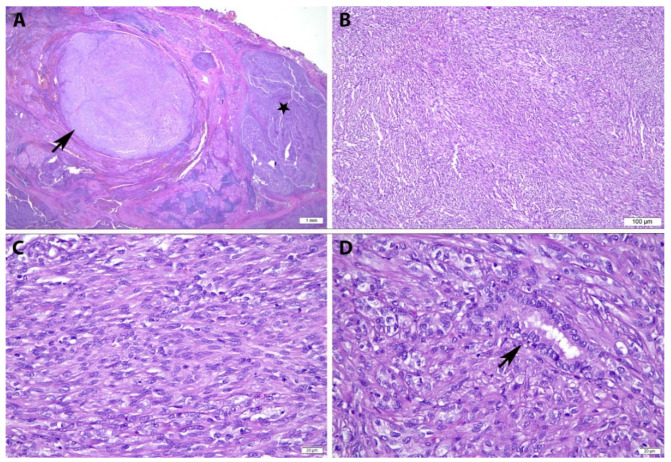
Histopathological Features of Papillary Thyroid Carcinoma with Spindle Cell Metaplasia. (**A**) A well circumscribed nodule (arrow) with adjacent tumour tissue displaying classical PTC (black star) (H&E, ×12.5); (**B**) The nodule comprises of spindle cells arranged in loose fascicles devoid of papillary architecture (H&E, ×100); (**C**) At a higher magnification, the spindle cells display uniform, plump to thin elongated nuclei with relatively abundant eosinophilic cytoplasm. No nuclear grooves and intranuclear cytoplasmic pseudoinclusions are found. No increase in mitosis nor necrosis is noted (H&E, ×400); (**D**) Entrapped tumour cells within the nodule exhibiting features consistent with classical PTC (arrow) are also observed (H&E, ×400).

**Figure 3 diagnostics-12-00855-f003:**
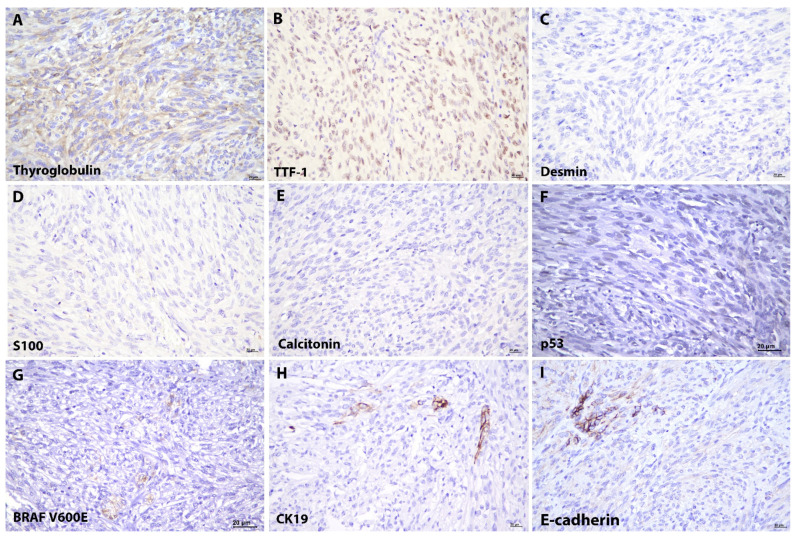
Immunohistochemical Features of Papillary Thyroid Carcinoma with Spindle Cell Metaplasia. The spindle cells exhibit (**A**) thyroglobulin (thyroglobulin, ×400) and (**B**) thyroid transcription factor-1 (TTF-1, ×400) immunopositivity, confirming the thyroid follicular cells in origin. They are immunonegative to (**C**) desmin (desmin, ×400), (**D**) S100 (S100, ×400) and (**E**) calcitonin (calcitonin, ×400). (**F**) p53 (p53, ×400) and (**G**) BRAF V600E mutant protein (BRAF, ×400) were focally expressed by the spindle cells. The spindle cells show a loss of (**H**) CK19 (CK19, ×400) and (**I**) E-cadherin (E-cadherin, ×400) immunoreactivity, compared to the entrapped classical PTC.

**Table 1 diagnostics-12-00855-t001:** Clinicopathological Features of Reported Cases on Papillary Thyroid Carcinoma with Spindle Cell Metaplasia.

No.	Authors	Age(Years)/Gender	Clinical Presentation	Size (mm)	Histopathological Subtype	Immunohistochemistry Staining Pattern	
						Follicular Cell Component	Spindle Cell Component
1.	Vergilio et al. (2002) [2]	25–61/6 F, 1 M	Mass at left/right thyroid lobe	3–30	Spindle cell variant (1%–95%)	NA	TG+, CK+/−, calcitonin−, SMA−
2.	Woenckhaus et al. (2004) [3]	32/F	Cold nodule over right thyroid lobe	27	Spindle cell variant (80%) + Follicular variant (20%)	TG+, CKAE1/AE3+, CAM5.2+, TTF-1+, vimentin+, S100+, BCL-2+, 34βE12+, E-cad+, PR+ *, ER−, calcitonin−, chromo−, synapto−, SMA−, CD34−, HMB45−, p53−	TG+, CKAE1/AE3+, CAM5.2+, TTF-1+, vimentin+, S100+, BCL-2+, 34βE12−, E-cad−, PR+ *, ER−, calcitonin−, chromo−, synapto−, SMA−, CD34−, HMB45−, p53−
3.	Rossi et al. (2012) [4]	51/M	Left thyroid nodule	50	Spindle cell variant (almost 100%)	NA	TG+, TTF-1+, HMBE-1+, galectin-3+, CK+, calcitonin−, CD34−, chromo−, synapto−, S100−, CEA−, p53−, Ki67 < 5%
4.	Corrado et al. (2013) [5]	56/M	Mass in left thyroid lobe	45	Spindle cell variant (80%) + Follicular variant (20%)	TTF-1+, CK19+	TG+, TTF-1+, MIB < 1%, CKMNF116−, CK19−, CK7−, p63−, chromo−, calcitonin−, S100−, NSE−, SMA−, myosin−, desmin−, EMA−, CD1a−, CD3−, CD5−, CD21−, CD31−, CD34−, CD35−, CD68−, CD117−, BCL-2−, melan A−, HMB45−, p53−
		53/F	Mass in right thyroid lobe	46	Spindle cell variant (80%) + Follicular variant (20%)	TTF-1+, CK Fil7+8+	TG+, TTF-1+, CKMNF116+, MIB < 2%, CK19−, CK7−, p63−, chromo−, calcitonin−, S100−, NSE−, SMA−, myosin−, desmin−, EMA−, CD1a−, CD3−, CD5−, CD21−, CD31−, CD34−, CD35−, CD68−, CD117−, BCL-2−, melan A−, HMB45−, p53−
5.	Ma et al. (2015) [6]	67/M	Mass in right thyroid	25	Spindle cell variant (almost 100%) with rare neoplastic follicular cells	TG+, TTF-1+, PAX-8+, vimentin+, BCL-2+, panCK+, CAM5.2+, CD34-, CD99−, CK7−, CK19−, ER−, PR−, calcitonin−, synapto−, chromo−, S100−, CD21−, CD23−, SMA−, HMB45−, p63−, p40−, p53−	TG+, TTF-1+, PAX-8+, vimentin+, BCL-2+, panCK−, CAM5.2−, CD34−, CD99−, CK7−, CK19−, ER−, PR−, calcitonin−, synapto−, chromo−, S100−, CD21−, CD23−, SMA−, HMB45−, p63−, p40−, p53−, Ki67 3%
6.	Present case (2021)	51/F	Cold nodule over left thyroid lobe	30	Classical type (70%) Spindle cell variant (30%)	TG+, TTF-1+, CKAE1/AE3+, CK19+, E-cad+, vimentin+, calcitonin−, S100−, desmin−, SMA−, chromo−, synapto−	TG+, TTF-1+, p53+, BRAF V600E+, CKAE1/AE3+, CK19−, E-cad−, vimentin+, calcitonin−, S100−, desmin−, SMA−, chromo−, synapto−

“+” = positive, “−” = negative, “*” = isolated nuclei positivity, chromo = chromogranin, CK = cytokeratin, E-cad = E-cadherin, ER = oestrogen receptor, F = female, M = male, NA = not available, NSE = neurone specific enolase, PR = progesterone receptor, SMA = smooth muscle actin, synapto = synaptophysin, TG = thyroglobulin, TTF-1 = thyroid transcription factor-1.

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
