# Peer review of "Papillary Thyroid Carcinoma with Spindle Cell Metaplasia: A Rare Encounter"

_diagnostics, 2022, doi:10.3390/diagnostics12040855_

Round 1

Reviewer 1 Report

Many histological variants of papillary thyroid carcinoma have been described and PTC with spindle cell metaplasia is rare one type.

This is a well written case report presenting a case of spindle cell metaplasia incidentally found in a PTC. This article gives a good overview of the literature and a good presentation of the case with nice images. 

The authors should better clarify the workup made on this patient and the surgical strategy, especially:

Report the Bethesda cytology classification of the lesion (B6?)

Why was associated a lateral neck dissection? only on the basis of US findings? does a fine needle biopsy was made on lateral lymph nodes to justify the lateral neck dissection ? Thyroglobulin dosage on it?

What was the histopathological result on central and lateral lymph node dissection? Does the same histologic architecture was present in lymph nodes (if metastatic?)

best regars

Author Response

Many histological variants of papillary thyroid carcinoma have been described and PTC with spindle cell metaplasia is rare one type.

This is a well written case report presenting a case of spindle cell metaplasia incidentally found in a PTC. This article gives a good overview of the literature and a good presentation of the case with nice images. 

Thank you for the kind comments.

The authors should better clarify the workup made on this patient and the surgical strategy, especially:

  1. Report the Bethesda cytology classification of the lesion (B6?)

Thank you for the comment. Yes, the lesion was reported under the Bethesda diagnostic category malignant (Bethesda category 6). The diagnostic category was added in the manuscript as suggested (see page 2, line 76).

  1. Why was associated a lateral neck dissection? only on the basis of US findings? does a fine needle biopsy was made on laterallymph nodes to justify the lateral neck dissection? Thyroglobulin dosage on it?

Total thyroidectomy and lateral neck dissection were performed following the cytopathological diagnosis of a papillary thyroid carcinoma (PTC). It is a local practice that in any cases of cytological confirmation of PTC, lateral neck node dissection will be carried out as part of the management of a PTC since lymph node metastases are a common occurrence in PTC. This was added in page 2, line 78 as suggested.

Fine needle aspiration biopsy was not made on the lateral lymph nodes in this patient.

Thyroglobulin that was mentioned in this manuscript was an antibody that we routinely use in the laboratory as an immunohistochemistry technique to identify thyroid origin of the tumour. Similar to thyroid transcription factor-1 (TTF-1), demonstration of thyroglobulin immunopositivity (as in this case) confirmed its thyroid in origin.

  1. What was the histopathological result on central and laterallymph node dissection? Does the same histologic architecture was present in lymph nodes (if metastatic?)

Thank you for the questions. Sections from the lymph nodes showed evidence of distance metastases from classic PTC. Yes, they displayed similar histologic morphology that was described in the thyroid mass. (see page 3, line 94-95)

We hope that we have had answered the reviewers’ comments and suggestions satisfactorily. Thank you.

Reviewer 2 Report

The authors reported a case of a female patient with a spindle cell variant of PTC, and subsequently elaborate on this type of thyroid cancer.

After reading the manuscript some important issues remain:

  1. General comments
  • In the Title, Abstract and Conclusion the term ‘one size does not fit all’ is used, but it is not clear to me what is meant by this. Please explain.
  1. Introduction

Line 45/46: less than 5% to more than 95%; I assume this is incorrect and should be ‘more than 5% to less than 95%’.

  1. Cytopathological and Histopathological Examination
  • Line 78-94. It seems to me that there are two separate node of which the largest (30mm) is a classical PTC and the smaller one (9mm) is a spindle cell variant. One might argue that the smaller node is just an incidental finding and it is questionable whether this would affect prognosis, while at the other hand the lymph node metastases of the classical PTC will do probably have an impact on prognosis.
  • Line 121. This probably should be pT2N1b instead of pT2N1.
  1. Discussion
  • Line 130-133. I would suggest to add ‘According to literature’, because e.g. you probably can find someone in the real world with a spindle cell variant being 68 years of age.

Author Response

The authors reported a case of a female patient with a spindle cell variant of PTC, and subsequently elaborate on this type of thyroid cancer.

After reading the manuscript some important issues remain:

  1. General comments

In the Title, Abstract and Conclusion the term ‘one size does not fit all’ is used, but it is not clear to me what is meant by this. Please explain.

Thank you for the comments. The term “one size does not fit all” is referred to the presence of spindle cells in a case of thyroid carcinoma. When encountering a thyroid nodule consisting of spindle cells, before one consider signing out a diagnosis of spindle cell variant of papillary carcinoma, other possible differential diagnoses such as smooth muscle tumours of thyroid, peripheral nerve sheath tumours of thyroid, spindle epithelial tumours with thymus-like differentiation (SETTLE) and etc need to be considered and excluded. Careful evaluation of histomorphological features and judicious use of immunohistochemistry stains may be useful in this context. Possible differential diagnosis and helpful diagnostic features were included in the discussion section.

Having said that, we are willing to remove the term “one size does not fit all” if the reviewer thinks that this is deemed unsuitable to be used in our manuscript.

  1. Introduction

Line 45/46: less than 5% to more than 95%; I assume this is incorrect and should be ‘more than 5% to less than 95%’.

Thank you for pointing that out. We had changed the statement to “from 1% to almost 100%” to make it clearer. (see page 2, line 46)

  1. Cytopathological and Histopathological Examination
  • Line 78-94. It seems to me that there are two separate node of which the largest (30mm) is a classical PTC and the smaller one (9mm) is a spindle cell variant. One might argue that the smaller node is just an incidental finding and it is questionable whether this would affect prognosis, while at the other hand the lymph node metastases of the classical PTC will do probably have an impact on prognosis.

Thank you for the comments. Yes, the spindle cell variant constitutes about 30% of the entire tumour bulk in this case. Whether or not it will be prognostically significant we are not sure at this point of time. Limited literature was previously published on the prognostic significance of this extremely rare variant. The impact on prognosis we believe will still fall on the presence of lymph node metastasis from classical PTC in this case. The patient was treated as such with radioactive iodine post operatively and was well at the time of writing.

  • Line 121. This probably should be pT2N1b instead of pT2N1.

Thank you for pointing that out. We had corrected it to pT2N1b as suggested. (see page 4, line 122)

  1. Discussion

Line 130-133. I would suggest to add ‘According to literature’, because e.g. you probably can find someone in the real world with a spindle cell variant being 68 years of age.

Thank you for the suggestion. We had added in “according to literature” as suggested with relevant references (see page 4, line 130).

We hope that we have had answered the reviewers’ comments and suggestions satisfactorily. Thank you.

Round 2

Reviewer 1 Report

thank you for the revisions my only concern is due to the fact that has been associated a lateral neck dissection that is not suggested by recommandations unless in preoperative US were detected suspect lymph nodes. So justify this attitude on the case report to avoid less experienced reader to think that this is the recommended practice for this kind of tumors. 

Author Response

Responses to Reviewer 1

Thank you for the revisions my only concern is due to the fact that has been associated a lateral neck dissection that is not suggested by recommendations unless in preoperative US were detected suspect lymph nodes. So justify this attitude on the case report to avoid less experienced reader to think that this is the recommended practice for this kind of tumors. 

Our responses:

Thank you for the comments. Yes, preoperative ultrasonography showed a few suspicious ipsilateral lymph nodes seen in this patient, hence lateral neck dissection was performed in addition to total thyroidectomy. Additional statements on the suspicious lymph nodes were added as suggested (see page 2, lines 64-66; and page 3, lines 80-81).

Reviewer 2 Report

I’m happy with changes the authors made in this revised manuscript.

Just a one issue remains:

  1. General comments
  • About ‘one size does not fit all’. After reading your explanation it is now clear to me what is meant by this. However this was not clear from the original text. Either, add your explanation to the discussion, or leave out ‘one size does not fill all’ (but then it should also be removed from the title).

Author Response

Responses to Reviewer 2

I’m happy with changes the authors made in this revised manuscript.

Just a one issue remains:

General comments

  • About ‘one size does not fit all’. After reading your explanation it is now clear to me what is meant by this. However this was not clear from the original text. Either, add your explanation to the discussion, or leave out ‘one size does not fill all’ (but then it should also be removed from the title).

Our responses:

Thank you for the comments. We agree to remove the term “one size does not fit all” from the manuscript (see page 1, line 27; page 7, line 223) as well as the title. Following that, a slight change in the title was also made (see page 1, lines 2-3).